# A Low-Power Electrothermal Flexible Actuator with Independent Heating Control for Programmable Shape Deformation

**DOI:** 10.3390/mi16040456

**Published:** 2025-04-11

**Authors:** Shen Dai, Zhiyao Ling, Han Gong, Kunwei Zheng

**Affiliations:** School of Optoelectronic and Communication Engineering, Xiamen University of Technology, Xiamen 361024, China; 2322101003@stu.xmut.edu.cn (S.D.); 2422121017@stu.xmut.edu.cn (Z.L.); 2422111007@stu.xmut.edu.cn (H.G.)

**Keywords:** flexible actuator, joule heating, programmable deformation, low-voltage actuation, thermal expansion

## Abstract

Flexible actuators hold significant promise for applications in intelligent robotics, wearable devices, and biomimetic systems. However, conventional actuators face challenges such as high driving voltages, inadequate deformation control, and limited deformation modes, which hinder complex programmable dynamic deformations. This study presents an electrothermal actuator based on a conductive silver paste/Kapton/PDMS composite structure, enabling precise and adjustable deformation through programmable thermal control. Experimental results show that the actuator achieves a large-angle bending (∼203°) within 12 s under a low driving voltage of 2.0 V. Compared to the PTFE/MXene/PI structure, the proposed actuator achieves a 64% increase in bending angle, a 70% reduction in response time, and a 67% decrease in driving voltage. By independently controlling multiple heating elements, the actuator exhibits programmable deformation modes, including local, symmetric, and sinusoidal bending. The relationship between input voltage and deformation amplitude is described using a sinusoidal function model, experimentally validated for accuracy. Compared to traditional actuators, the proposed design offers significant improvements in bending angle, response speed, and voltage requirements. By optimizing the conductive silver paste pattern and voltage input strategy, this work develops a low-voltage, highly controllable, multi-mode programmable actuator with potential for applications in flexible robotics and space-deformable antennas.

## 1. Introduction

Flexible actuators have attracted significant attention in fields such as intelligent robotics, wearable devices, bionic systems, and reconfigurable electronic devices. Compared to traditional rigid actuators, flexible actuators are lightweight, highly flexible, and deformable, enabling them to adapt to complex environments and exhibit multiple deformation modes [1]. However, current flexible actuators often lack programmability, exhibit a single deformation mode, and face challenges in achieving complex programmable deformations. Furthermore, many flexible actuators require high-voltage drives (above 6 V), and their deformation response times generally exceed 40 s, which limits their applicability in low-power systems.

Researchers have been exploring various driving methods, including optical driving [2], electrical driving [3], thermal driving [4], magnetic field driving [5], chemical driving [6], and humidity-responsive driving [7]. Among these, electrical stimulation has emerged as a particularly promising driving mechanism due to its fast response, ease of control, and high programmability. Electrical stimulation can be classified into two primary activation mechanisms: field activation and current activation [8]. Field activation typically requires high-voltage drives (>1 kV), while current activation utilizes the Joule heating effect to induce thermal expansion in materials, enabling low-voltage, high-bending deformation. This makes current activation well-suited for low-power flexible actuators [9].

Recent advancements in electrothermal flexible actuators, based on various materials, have demonstrated significant progress; however, challenges persist in terms of slow response times, high driving voltages, and limited deformation angles. For example, Sang et al. proposed a “cut-and-paste” assembly method for PTFE/MXene/PI composite structures, achieving a maximum bending angle of 122° in 40 s at a low voltage of 6 V [10]. Li et al. developed a Nafion/Ag composite ultrafine fiber film, which exhibited a bending angle of 118° at 2.4 V [11]. Nguyen et al. employed photolithography and etching techniques to fabricate a PDMS/p-Cu/PI structure, which attained a bending angle of 160° in 10 s at 2.25 V [12]. Additionally, Tian et al. developed a PE/carbon nanocomposite actuator that reached a bending angle of 221° within 20 s at 5 V [13]. While these studies have made strides in reducing driving voltage or increasing bending angles, most solutions still require higher voltages or longer response times. Additionally, the programmability of their deformation modes remains limited, complicating precise control over bending morphology. Therefore, developing flexible actuators with low power consumption, high response speed, substantial bending angles, and programmable control remains a key research focus.

This study presents an electrothermal actuator based on a conductive silver paste/Kapton/PDMS composite structure, designed to address the issues of single deformation modes, slow response speeds, and high driving voltages in existing flexible actuators. Compared with conventional electrothermal drive structures (e.g., PTFE/MXene/PI [10]), the proposed actuator exhibits superior processability and structural programmability, owing to its enhanced material availability, improved interfacial stability, and precise patterning controllability. The proposed actuator achieves precise deformation control through programmable heating. Experimental results demonstrate that the actuator can achieve controlled bending deformation of approximately 203° in 12 s at a low voltage of 2.0 V. Compared to the PTFE/MXene/PI structure, the proposed actuator shows a 64% increase in bending amplitude, a 70% improvement in response speed, and a 67% reduction in driving voltage [10]. By independently controlling multiple heating components, the actuator can exhibit various deformation modes, such as local bending, symmetrical bending, and sinusoidal bending, offering greater flexibility and programmability for applications in flexible robotics and reconfigurable electronic devices. A sinusoidal function model is employed to quantitatively describe the relationship between input voltage and deformation amplitude. The model’s accuracy is validated experimentally, enabling precise control over deformation. By optimizing the pattern design and input voltage of the conductive silver paste, the actuator can achieve lower driving voltages, faster response times, higher bending angles, and precise control of complex deformations. This actuator holds substantial potential for applications in flexible robotics, space-deformable antennas, and other fields.

## 2. Experimental Section

### 2.1. Materials

Conductive silver paste was purchased from Yilai Technology Co., Ltd. (Shenzhen, China) and used as received without further purification. The polydimethylsiloxane (PDMS) precursor and curing agent (Sylgard 184) were obtained from Dow Corning (Midland, MI, USA) and were also used as received without additional purification. The polyimide film (Kapton, DuPont, Wilmington, DE, USA) used in this study had a thickness of 50 μm and was incorporated with an organic silicone adhesive. All these reagents were used without further purification.

### 2.2. Characterization

The voltage was applied using a direct current source (GER916307, GW Instek, Suzhou, China). The temperature of the device was measured by using a non-contact thermometer (Fotric 246M, Shanghai, China).

### 2.3. PDMS Film Preparation

To prepare the PDMS film, the PDMS prepolymer and cross-linking agent were mixed thoroughly in a 10:1 weight ratio to form a uniform mixture. The mixture was degassed under a vacuum to remove air bubbles formed during the mixing process, ensuring that the film had no significant pore defects. The resulting mixture was then precisely printed onto a PET substrate using a multifunctional flexible electronic printing device, allowing for accurate control of the film’s thickness and morphology. The printed PDMS film was subsequently cured at 90 °C for 20 min on a heating table in the flexible electronic printing device to achieve the desired mechanical properties.

## 3. Results

### 3.1. Design, Electro Thermal Mechanism, and Fabrication of Flexible Actuator

The flexible actuator designed in this study features a simple structure, composed of three primary layers: silver paste, Kapton, and PDMS, as shown in Figure 1a. PDMS, an organic silicon polymer with a silicon–oxygen backbone, was selected due to its excellent mechanical flexibility, chemical stability, biocompatibility, and high thermal expansion coefficient of $310\times 10^{-6}/{}^{\circ}C$, which allows for significant thermal expansion at low temperatures [14]. The Kapton film serves as the intermediate structural layer, offering superior mechanical flexibility, low thermal expansion, good physical adhesion, and a low friction coefficient, making it an ideal material for this application [15]. When combined, the differing thermal expansion coefficients of PDMS and Kapton enable rapid bending and deformation in response to temperature changes. Silver paste, acting as the conductive heating layer, has a square resistance as low as 60 mΩ/sq, ensuring high conductivity, fast heating, and good flexibility. Upon activation, the silver paste layer induces bending toward the Kapton side through the Joule heating effect. When the power is turned off, the actuator cools and returns to its initial flat state, demonstrating reversible, repeatable bending behavior, as shown in Figure 1b. Figure 1c outlines the fabrication process of the flexible actuator, which utilizes a multifunctional flexible printing device. The process consists of four main steps: (1) A PET film is used as a flexible printing substrate, and the PDMS mixed solution is printed onto the PET film, followed by heating at 90 °C for 20 min; (2) a Kapton film is then applied to the PDMS surface; (3) silver paste patterns are printed onto the PDMS and Kapton layers using the flexible printing device and cured at 90 °C for 30 min; (4) finally, copper wires are connected to the silver paste layer to complete the flexible actuator.

### 3.2. Electro Thermal Performance and Actuation Characteristic Sofa Flexible Actuator

In the design of flexible heaters, electrothermal performance is a critical factor. Commonly used heating materials include carbon nanotubes [16], graphene [17], silver nanoparticles [18], PEDOT:PSS [19], and conductive silver paste [20], among others. While carbon nanotubes and graphene exhibit excellent electrical conductivity, their relatively high resistance necessitates higher voltages to achieve effective heating during the electrothermal conversion process. This requirement can pose limitations for low-voltage drive applications. In contrast, silver paste offers lower resistivity, allowing for effective heating at lower driving voltages. Therefore, for flexible actuators, particularly in applications that require low-voltage driving and high-efficiency heating, conductive silver paste stands out as an ideal electrothermal material due to its superior electrothermal performance.

The electric driving performance of the flexible actuator, particularly its bending deformation ability and dynamic response speed, is primarily determined by the thermal properties of the flexible electric heating film. Key parameters, such as the uniformity of the temperature field distribution, heating rate, and steady-state temperature, play a decisive role in actuator performance. In this study, a flexible actuator based on a conductive silver paste/Kapton/PDMS composite structure was designed and developed. The sample dimensions are 55 mm × 10 mm × 0.35 mm, featuring a multilayer architecture consisting of a 0.2 mm-thick PDMS substrate, a 50 μm-thick Kapton film, and a 0.1 mm-thick silver paste conductive layer. The device exhibits an electrical resistance of 9.4 Ω, and its optical image is shown in Figure 2a. Upon applying a 1.5 V driving voltage, the conductive silver paste layer generates heat through the Joule effect, which is transferred to the PDMS layer via the interface. Due to the significant difference in thermal expansion coefficients between the Kapton film and PDMS, the PDMS layer undergoes anisotropic thermal expansion, causing the actuator to bend toward the Kapton film side, as shown in Figure 2b. The surface temperature distribution of the device was monitored using a thermal imager (Figure 2c). After power is turned off, the heat dissipates, and the device returns to its original flat state.

To further investigate the electrothermal properties of the conductive silver paste, the electrothermal response characteristics of the sample device were measured under various driving voltages. Figure 2d shows the maximum temperature change curve of the device as the voltage is varied from 1.0 to 3.0 V. The results from three repeated experiments indicate that, as the driving voltage increased from 1.0 to 3.0 V, the device temperature rises from 55.7±1.3°C to 227±2.1°C. The inflection point in the temperature profile (Figure 2d) marks the cessation of device bending deformation, coinciding with automatic power supply disconnection. When the power is turned off, the temperature of the device decreases and the initial shape is restored. Linear regression analysis demonstrates a strong linear correlation between input voltage and current. The relationship is well described by the equation I=0.61(Ω)V, where *I* is the current in amperes (A), and *V* is the voltage in volts (V). The coefficient of determination ($R^{2}$ = 0.997) indicates an excellent fit to the experimental data, as shown in Figure 2e. This result suggests that increasing input power raises the saturation temperature of the device [23]. The surface temperature of the device can be precisely adjusted through an external voltage. As shown in Figure 2f, significant differences exist in the bending angles and driving voltages among various materials. For example, the PI/LRGO/Ag composite requires a driving voltage of 28 V to achieve a bending angle of only 192°, whereas the material developed in this work achieves a bending angle of 203° at just 2 V, demonstrating the excellent electrothermal properties of silver paste at low voltages. Moreover, compared with the electrothermal drivers reported in the literature (see Table 1), the drivers presented in this study exhibit superior performance in terms of driving voltage, bending angle, and response time. The actuator achieves rapid bending deformation at low driving voltages, outperforming comparable devices. These experimental findings indicate that the driving voltage directly influences the heating rate, saturation temperature, and, consequently, the bending deformation angle of the actuator. The flexible actuator in this study demonstrates outstanding electrothermal performance under low-voltage driving, showing significant potential for future low-power flexible actuator designs.

### 3.3. Electro Thermal Bending Behavior of Flexible Actuator Under Varying Voltages

Figure 3a illustrates the response characteristics of the sample device driven by a 2.0 V DC voltage. Upon applying the voltage, the conductive silver paste layer converts electrical energy into heat through the Joule effect, which is then transferred to the PDMS layer. This causes anisotropic molecular expansion within the material, resulting in volume expansion of the PDMS. Since the thermal expansion coefficient of PDMS is significantly larger than that of the Kapton film, the PDMS expands more substantially as the temperature rises, leading to the actuator bending toward the Kapton side. When the power is disconnected, the heat dissipates gradually, the temperature decreases, and the actuator returns to its initial flat state. Experimental results indicate that, under a 2.0 V DC voltage, the actuator reaches a bending angle of 180 ± 3.2° or more within 10 s. After disconnecting the power at 12 s, the actuator returns to its original state after approximately 17 s, as shown in Figure 3a. To further investigate the impact of driving voltage on the bending angle, the actuator’s deformation was tested under 1.5, 2.0, and 2.5 V DC voltages, as shown in Figure 3b. The results demonstrate that, as the driving voltage increases from 1.5 to 2.5 V, the bending angle of the actuator progressively increases, the saturation temperature rises, and the time required to reach the maximum bending angle decreases significantly. To analyze the temperature change in more detail, a thermal imager was employed to measure the surface temperature of the actuator at voltages ranging from 1.0 to 3.0 V, as shown in Figure 3c. The data show that, as the voltage increases, the surface temperature of the actuator rises. At a 1.0 V DC voltage, the bending deformation is minimal, and the maximum surface temperature reaches 57.9±1.1°C. At 1.5 V, the maximum temperature reaches 90.5±1.6°C, corresponding to a bending angle of approximately 77.9 ± 2.4°. At 2.0 V, the bending angle increases to about 203 ± 3.5°, with a maximum surface temperature of 150.1±2.3°C. At 2.5 V, the maximum surface temperature rises to 197.2±2.7°C, and the bending angle increases further. Under a 3.0 V voltage, the maximum surface temperature reaches 227.0±2.8°C, and the actuator achieves maximum bending deformation. These experiments show that, as the driving voltage increases, the Joule heating effect intensifies, leading to an increase in both the bending angle and response speed. The cooling recovery time also increases, indicating that the heat dissipation process becomes slower at higher temperatures. Overall, the bending angle of the actuator is primarily controlled by the applied driving voltage and increases with increasing voltage.

### 3.4. Programmable Flexible Actuators with Independent Heating and Controlled Bending Deformation

To achieve programmable control of the actuator shape, this study introduces independent heating components into the device structure and realizes multiple deformation modes by adjusting the placement of the Kapton film. Figure 4a presents a schematic illustration of the structure, featuring two independent silver paste heater components, where the heaters at both ends of the device can be individually controlled. Upon applying electrical stimulation, the originally flat structure undergoes controllable bending deformation depending on the heating method. The controllability of the bending direction primarily depends on the placement of the Kapton film. Since Kapton has a low thermal expansion coefficient, its presence constrains the thermal expansion direction of the PDMS, thereby determining the bending trend of the device. By adjusting the arrangement of the conductive silver paste and the Kapton film within the PDMS layer, programmable directional bending control can be achieved at low driving voltages. To validate this concept, two representative deformation modes were selected for experimental verification. During the fabrication process, a multifunctional flexible printing device was used to manufacture a conductive silver paste heating component (15mm×5mm×0.1mm) and a Kapton film (20mm×10mm×0.05mm). The heating components were attached to the upper and lower surfaces of the 0.2 mm-thick PDMS, with the Kapton film covering the silver paste layer to regulate the bending direction. Figure 4a(i) illustrates the structural schematic, and Figure 4b(i) presents the optical image of the actuator in its initial state. In experimental tests, DC voltages of 2.3 V and 1.7 V were applied to the upper and lower heating components, respectively, causing the device to exhibit a waveform similar to a sine function, as depicted in Figure 4c. To quantitatively describe the relationship between voltage and amplitude, a sine function model was employed for curve fitting. When DC voltages of 1.5 and 0.7 V were applied to the upper and lower heating components, the actuator produced small bending deformation with an amplitude of 5.125±0.12mm (Figure 4d). Increasing the voltage to 2.0 and 1.0 V significantly enhanced the bending angle, resulting in an amplitude of 5.625±0.14mm (Figure 4e). Further increasing the voltage to 2.0 and 1.5 V led to a greater bending angle, with the amplitude reaching 5.95±0.15mm (Figure 4f). The experimental results indicate that the amplitude is a function of the applied voltage *V* and can be approximated using the following sine function:$$y\left(x\right)=A\left(V\right)\sin\left(\omega x\right)$$
where:$$\omega =\frac{2\pi }{T},\;A\left(V\right)=amplitude,\;T=2L$$
where 2*L* represents the bending cycle length of the actuator. The experimental findings reveal that, as the input voltage gradually increases, the actuator’s bending angle increases significantly, ultimately forming a large-angle sinusoidal bending deformation. This demonstrates that the driving voltage has strong controllability over the actuator’s deformation behavior. By applying different voltage combinations, the bending shape transformation of the device was analyzed, and the results were plotted in Figure 5. The data clearly indicate that, as the driving voltage increases, the bending amplitude also increases, further verifying the regulatory effect of driving voltage on actuator deformation.

To achieve programmable control of the actuator’s shape, this study further investigated the regulatory effect of integrating independent heating components at the left and right ends of the PDMS upper surface on the device’s deformation. As illustrated in Figure 4a(ii), a conductive silver paste heating component with dimensions 10mm×5mm×0.1mm was fabricated using multifunctional flexible printing technology and symmetrically positioned on both sides of the 0.2-mm-thick PDMS substrate. A Kapton film (15mm×10mm×0.05mm) was then applied to constrain the thermal expansion direction. The initial state of the device is shown in Figure 4b(ii). In experimental tests, when an average voltage of 0.95 V was applied, both ends of the device exhibited symmetrical bending deformation, reaching a warping height of 3.5±0.08mm at both ends and 1.5±0.06mm at the central region (Figure 4g). Increasing the voltage to 1.25 V resulted in a significant increase in bending amplitude, with the warping height reaching 4.5±0.10mm (Figure 4h). When the voltage is further increased to 1.45 V, the warping height at both ends is increased to 5±0.12mm, as shown in Figure 4i, 5.5±0.10mm at 1.6 V, as shown in Figure 4j, and 6±0.13mm at 1.9 V, as shown in Figure 4k. It is worth noting that, when the voltage is increased to 2.5 V, the warping height is further increased to 7±0.15mm, as shown in Figure 4l, indicating that the deformation amplitude is positively correlated with the driving voltage. To quantitatively analyze the relationship between applied voltage and deformation amplitude, the experimental data were linearly fitted, and the results are presented in Figure 6. Using the least squares method, the best-fitting equation was obtained:$$A\left(V_{\mathrm{DC}}\right)=KV_{\mathrm{DC}}+B$$
where $A(V_{\mathrm{DC}})$ represents the displacement height of the device, $V_{\mathrm{DC}}$ is the applied average voltage, K = 2.207 mm/V, B = 1.701 mm, and the determination coefficient $R^{2}=0.98$, indicating a high degree of consistency between the model and experimental data. For instance, when the input voltage was $V_{\mathrm{DC}}=0.95\;V$, the predicted deformation amplitude was $A(V_{\mathrm{DC}})=3.8\;\mathrm{mm}$, which closely matched the measured value of 3.5 mm. Similary, for $V_{\mathrm{DC}}=2.5\;V$, the model predicted $A(V_{\mathrm{DC}})=7.22\;\mathrm{mm}$, which was in close agreement with the measured value of 7.0 mm, further verifying the accuracy and reliability of the model.

The experimental results demonstrate that, as the input voltage increases gradually, the bending angle of the actuator increases significantly, ultimately forming a large-angle “S”-shaped bend. This indicates that the driving voltage has a controllable effect on the deformation of the device. By adjusting the on/off state of the heating components, reversible conversion between different shapes can be achieved. Furthermore, by adjusting the position and voltage of the heating components, the bending shape of the actuator can be precisely controlled. This provides a new strategy for the programmable deformation design of flexible actuators. Additionally, the combination of cutting processes and flexible printing technology can be employed to optimize the geometric shape of the device, enhancing its deformation controllability and environmental adaptability. This method not only improves the programmability of the device but also offers a novel approach for designing customizable flexible actuators in the future.

Conductive silver paste, known for its excellent electrical conductivity, can be precisely patterned using multifunctional flexible printing equipment, making it an ideal material for enhancing the deformation programmability of flexible actuators. Leveraging this characteristic, this study designed and fabricated a variety of irregularly shaped flexible actuators using flexible printing equipment. The basic structure of these actuators consists of conductive silver paste, a Kapton film, and PDMS arranged sequentially. In this setup, the conductive silver paste serves as the electric heat source to drive the actuator through the Joule effect, the Kapton film restricts the bending direction, and the PDMS acts as the flexible substrate providing mechanical support and deformation ability. The first actuator design studied is a ”petal-shaped structure” actuator with dimensions of 53.3mm×53.3mm×0.35mm, consisting of a 0.2 mm-thick PDMS substrate, a 50 μm-thick Kapton film, and a 0.1 mm-thick silver paste conductive layer. The optical images of the actuator are shown in Figure 7a(i,ii). When a 6.0 V DC voltage is applied, the conductive silver paste layer generates heat, causing each ”petal” to tilt upward synchronously, resulting in significant deformation, as shown in Figure 7a(iii,iv). This petal-like deformation drives the central region of the structure, where the antenna is located, to bend upward as well. Such coordinated movement may influence the spatial orientation and effective radiation pattern of the antenna, potentially offering opportunities for adaptive or reconfigurable communication applications. As shown in the thermal characteristics results (Figure 7a(v)), the triangle region exhibits the highest temperature. Point Sp1 indicates the highest local temperature, excluding the temperature along the corresponding conductive trace. Experimental results indicate that it takes approximately 15 s for the actuator to completely bend from its initial flat state under this voltage. The time required for this deformation depends primarily on material properties such as thermal conductivity and specific heat capacity, as well as the power input and geometric parameters of the device. In addition to keeping the original shape unchanged, the silver paste pattern was optimized, and different flexible actuators were designed to study their effects on deformation. After adjusting the silver paste pattern, a new actuator was prepared, with its structure shown in Figure 7b(i,ii). When a 5.0 V DC voltage was applied, significant bending occurred, and the results are displayed in Figure 7b(iii,iv). As shown in the thermal characterization results (Figure 7b(v)), point Sp1 shows the highest local temperature except for the corresponding conductive trace. In this case, the actuator’s design had been modified to feature four petals. The device showed precise and controllable deformation at DC voltages of 5.4 and 6.3 V, with the bending effects shown in Figure 7c(iii,iv). As shown in the thermal characteristics results (Figure 7c(v)), the triangle region exhibits the highest temperature. Point Sp1 indicates the highest local temperature, excluding the temperature along the corresponding conductive trace. The internal pattern of the silver paste was adjusted again to prepare another new actuator, as shown in Figure 7d(i,ii). This actuator was tested at 3.5 and 4.6 V DC voltages, and the corresponding bending effects are shown in Figure 7d(iii,iv). As shown in the thermal characteristics results (Figure 7d(v)), the triangle region exhibits the highest temperature. Point Sp1 indicates the highest local temperature, excluding the temperature along the corresponding conductive trace. Experimental findings show that adjusting the input voltage can effectively control the response speed and the maximum deformation angle of the actuator. By optimizing the arrangement of the silver paste, the deformation’s accuracy and programmability can be further improved. As the input voltage increases, both the heating rate and the maximum deformation angle of the actuator significantly improve. However, the cooling recovery time also increases, indicating that the actuator’s deformation is not only influenced by electrothermal conversion efficiency but is also constrained by the heat dissipation mechanism. In conclusion, this study demonstrates that, by optimizing the silver paste pattern and input voltage, the deformation mode of flexible actuators can be precisely controlled. This research provides a new strategy for the design of flexible programmable actuators, offering potential applications in fields such as soft robotics and bionic intelligent systems.

## 4. Conclusions

In this study, an electrothermal-responsive flexible actuator based on a conductive silver paste/Kapton/PDMS composite structure was developed, enabling precise deformation control through programmable heating. Experimental results demonstrate that, under a low driving voltage of 2.0 V, the actuator achieves a bending angle of approximately 203° within 12 s, with a maximum temperature reaching around 150.1 °C, showcasing excellent low-voltage driving and high-temperature response characteristics. Compared to existing electrothermal flexible actuators, the proposed actuator outperforms the PTFE/MXene/PI configuration in terms of bending speed, driving voltage, and bending angle. Specifically, the bending angle increased by 64%, the response time improved by 70%, and the driving voltage was reduced by 67%. Additionally, this study explored a programmable control strategy involving multiple heating components. By independently controlling these components, various deformation modes, such as local bending, symmetrical bending, and sinusoidal bending, were successfully achieved. To quantitatively analyze the relationship between input voltage and deformation amplitude, a sinusoidal function model was introduced for fitting. Experimental verification confirmed that the model effectively predicts the actuator’s deformation trend under different voltage conditions, further enhancing the actuator’s programmable control capabilities. This study highlights the potential of programmable heating control in flexible actuators and offers new design strategies for achieving low-voltage, high-reliability actuation. Although high-response bending was successfully realized under low-voltage conditions, the device may encounter thermal accumulation issues during high-temperature operation, particularly under continuous or repeated actuation scenarios. Future work may address these challenges by introducing thermally conductive channel structures, applying heat-resistant coatings, or implementing microstructural design strategies. Specifically, optimizing the microscale patterning of the heating layer or flexible substrate could enhance heat dissipation and long-term operational stability. These improvements would further expand the practical applicability of this technology in fields such as flexible robotics and space-deformable antennas.

## Figures and Tables

**Figure 1 micromachines-16-00456-f001:**
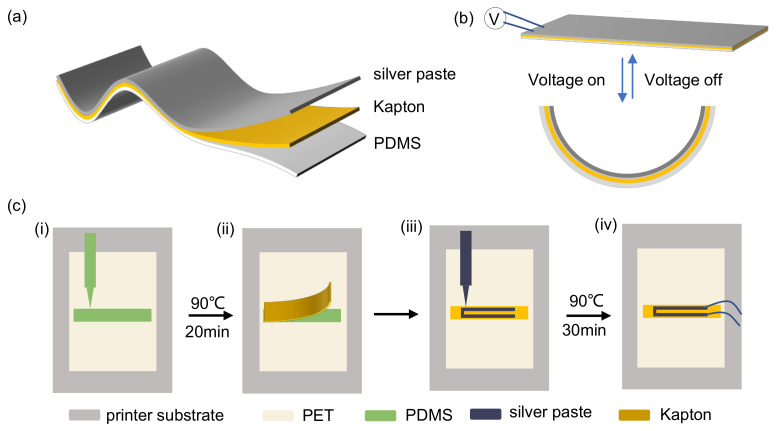
Structural and fabrication aspects of the flexible actuator. (**a**) Schematic illustration of the actuator structure; (**b**) electrothermal actuation mechanism; (**c**) fabrication process.

**Figure 2 micromachines-16-00456-f002:**
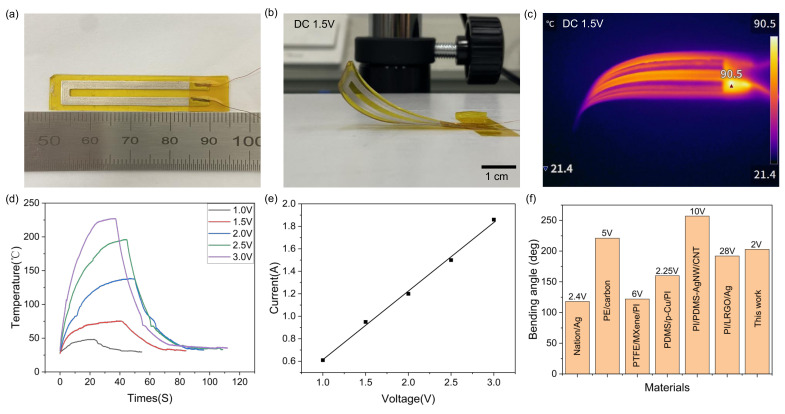
Electrothermal performance of the silver paste/Kapton/PDMS actuator. (**a**) Optical image of the actuator; (**b**) optical image of the actuator bending at 1.5 V; (**c**) temperature distribution image; (**d**) dependence of maximum device temperature on applied voltage; (**e**) linear relationship between input voltage and current. The solid line represents the linear fit to the data, with the equation I=0.61(Ω)V. (**f**) Comparative analysis of bending angle versus applied voltage for various actuator materials [10,11,12,13,21,22].

**Figure 3 micromachines-16-00456-f003:**
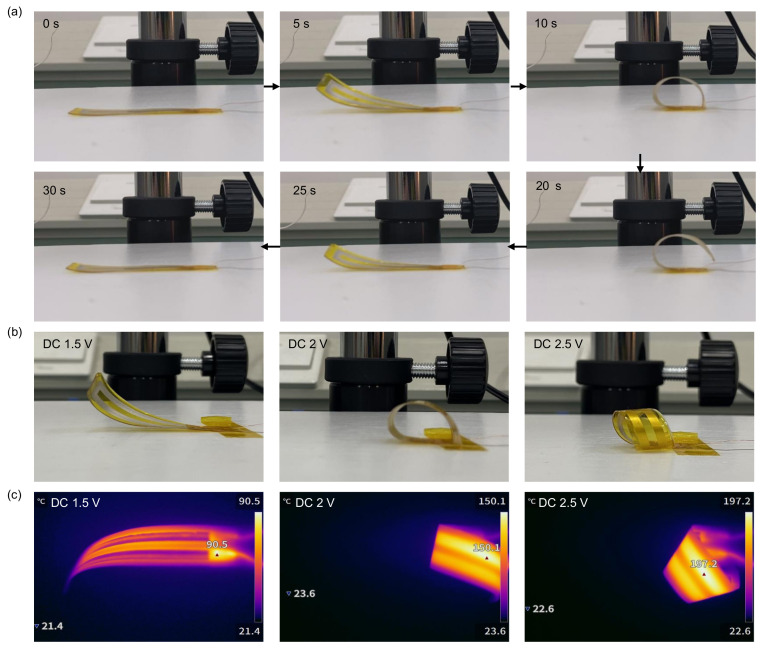
Electro thermal and deformation response of the actuator under different voltages. (**a**) Time-dependent optical images of the actuator at 2.0 V, with power off at 12 s; (**b**) optical images of the actuator at different applied voltages; (**c**) temperature distribution of the actuator under different voltages.

**Figure 4 micromachines-16-00456-f004:**
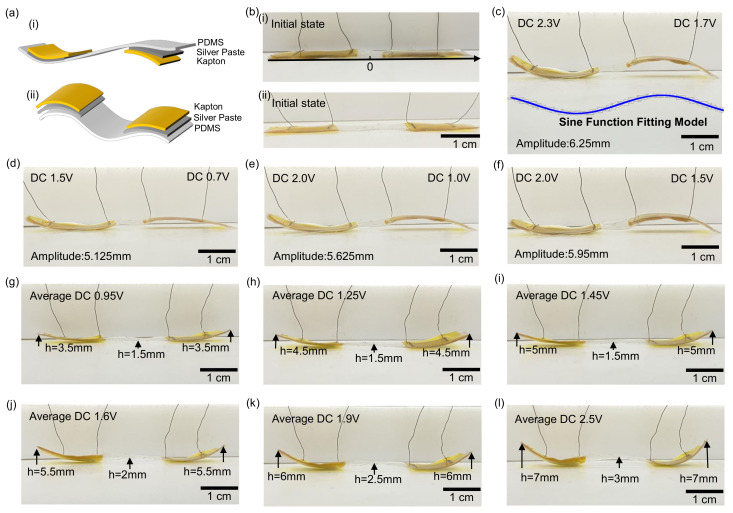
Programmable local bending response of the actuator under different voltages. (**a**) schematic illustration of the actuator structure: (i) structure with the heating layer on the opposite side, (ii) structure with the heating layer on the same side. (**b**) initial state of the actuator: (i) structure with the heating layer on the opposite side, (ii) structure with the heating layer on the same side. (**c**) comparison of bending response at 2.3 and 1.7 V with the sine function model; (**d**) bending response under 1.5 and 0.7 V; (**e**) bending response under 2.0 and 1.0 V; (**f**) bending response under 2.0 and 1.5 V; (**g**) bending response at an average applied DC voltage of 0.95 V; (**h**) bending response at an average applied DC voltage of 1.25 V; (**i**) bending response at an average applied DC voltage of 1.45 V; (**j**) bending response at an average applied DC voltage of 1.6 V; (**k**) bending response at an average applied DC voltage of 1.9 V. (**l**) bending response at an average applied DC voltage of 2.5 V.

**Figure 5 micromachines-16-00456-f005:**
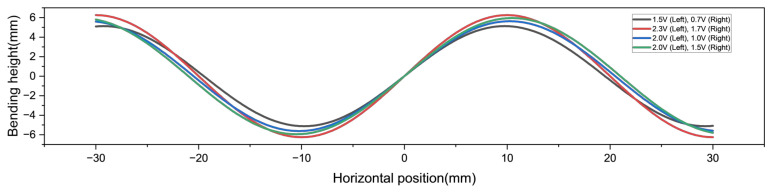
Voltage-controlled bending functions under four voltage combinations.

**Figure 6 micromachines-16-00456-f006:**
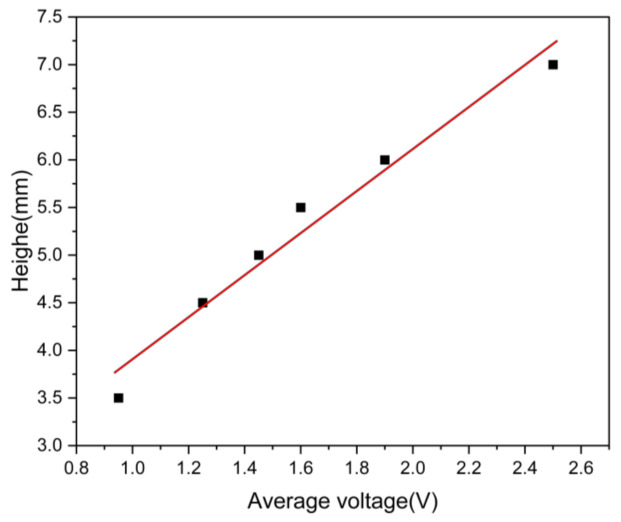
Experimental data and fitting curves of voltage and bending height. The solid line represents the linear fit to the data, with the equation $A\left(V_{\mathrm{DC}}\right)=2.207V_{\mathrm{DC}}+1.701$.

**Figure 7 micromachines-16-00456-f007:**
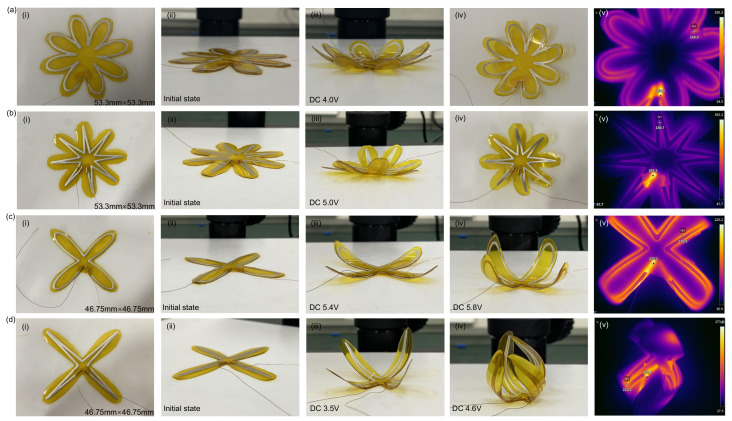
3D programmable bending deformation of the patterned actuator. (**a**) 3D deformation of the ”petal-like structure” actuator: (i) optical image of the device, (ii) initial state, (iii, iv) bending deformation under a DC voltage of 4.0 V, (v) temperature distribution under 4.0 V; (**b**) 3D deformation of different patterned ”petal-like structures”: (i) optical image of the device, (ii) initial state, (iii, iv) bending deformation under a DC voltage of 5.0 V, (v) temperature distribution under 5.0 V; (**c**) 3D deformation of the four-petal structure: (i) optical image of the device, (ii) initial state, (iii) bending deformation under a DC voltage of 5.4 V, (iv) bending deformation under 5.8 V, (v) temperature distribution under 5.8 V; (**d**) 3D deformation of different patterned four-petal structures: (i) optical image of the device, (ii) initial state; (iii) bending deformation under a DC voltage of 3.5 V, (iv) bending deformation under 4.6 V, (v) temperature distribution under 4.6 V.

**Table 1 micromachines-16-00456-t001:** Comparative of bending angle, driving voltage, response time, and operating temperature for flexible electrothermal actuators.

Materials	Voltage (V)	Bending Angle (deg)	Response Time (s)	Bending Rate (deg/s)	Temperature (°C)	Refs.
PTFE/MXene/PI	6	122	40	3.05	110	[10]
Nafion/Ag	2.4	118	12	9.83	66.8	[11]
PDMS/p-Cu/PI	2.25	160	10	16	160	[12]
PE/carbon	5	221	20	11.05	45	[13]
PI/PDMS-AgNW/CNT	10	257	80	3.21	139	[21]
PI/LRGO/Ag	28	192	6	31	78	[22]
This work	2	203	12	16.9	150.1	–

## Data Availability

The data presented in this study are available on request from the corresponding author.

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
