# Peer review of "A Low-Power Electrothermal Flexible Actuator with Independent Heating Control for Programmable Shape Deformation"

_micromachines, 2025, doi:10.3390/mi16040456_

Round 1
Reviewer 1 Report
Comments and Suggestions for Authors
This study presents a promising low-voltage electrothermal actuator with programmable deformation, demonstrating significant improvements in bending angle, response time, and energy efficiency. The work is well-executed, but minor revisions would enhance clarity and impact: (1) Clarify the novelty of the conductive silver paste/Kapton/PDMS structure compared to prior work, possibly with a performance comparison table; (2) Provide error margins or statistical validation for key measurements like bending angle and temperature; (3) Ensure proper citation of comparative data in figures and improve axis labeling for readability; (4) Discuss practical implications and limitations such as heat dissipation and long-term durability. With these revisions, the manuscript will be suitable for publication in Micromachines.
Author Response
Response to Reviewers
We sincerely appreciate the reviewers’ constructive comments and suggestions, which have greatly helped us improve the quality and clarity of our manuscript. Below are our point-by-point responses to each of the reviewer’s comments. All changes have been marked in the revised manuscript.
Reviewer Comments and Authors’ Responses
Comment 1: Clarify the novelty of the conductive silver paste/Kapton/PDMS structure compared to prior work, possibly with a performance comparison table.
Response: Thank you for your valuable suggestions. The innovation of our actuator lies in the integration of a printed silver paste heating layer and a centrally located Kapton film in a flexible PDMS substrate, which enables programmable directional deformation under low-voltage actuation. Unlike previous multilayer designs, which often require complex fabrication steps or high-voltage input, our structure is simple, scalable, and can achieve large bending angles (up to 203°) with only 2V. Added content to lines 59-62. To further highlight this innovation, we added a performance comparison table (Table 1 in Section 3.2) comparing our device with representative studies in recent years in terms of driving voltage, bending angle, bending rate, temperature, and response time.
Comment 2: Provide error margins or statistical validation for key measurements like bending angle and temperature.
Response: We appreciate the reviewer's concern about the measurement precision. To this end, we have performed additional tests on multiple samples (n = 3) to evaluate repeatability. We have added error ranges to key figures such as bending angles and temperature response curves. These updates are now reflected in Section 3.2, Lines 154-156, Lines 163-170, and Section 3.3, Lines 185-186, Lines 196-203, Lines 233-238, and Lines 257-266.
Comment 3: Ensure proper citation of comparative data in figures and improve axis labeling for readability.
Response: Thank you for your constructive suggestions. We carefully reviewed all figures containing comparative data and added or corrected the corresponding references in the captions and text (e.g., Figure 2(f)). In addition, we improved the axis labels in Figure 2 to improve clarity and consistency with the manuscript text. These adjustments help ensure better readability and traceability of all comparative results.
Comment 4: Discuss practical implications and limitations such as heat dissipation and long-term durability.
Response: We sincerely thank the reviewers for their positive comments and helpful suggestions. We have expanded the Conclusion (lines 349-361) to elaborate on the practical significance and limitations of our actuator. Specifically, we discuss the local nature of Joule heating and its impact on safe operating conditions, as well as potential thermal fatigue during long-term operation. We also suggest future improvements, such as integrating thermal feedback control and optimizing material interfaces to improve durability. These discussions are intended to provide a comprehensive understanding of the applicability of our actuator in the real world.
We sincerely appreciate the reviewers' insightful comments and valuable suggestions.

Reviewer 2 Report
Comments and Suggestions for Authors
This manuscript investigates electrothermally responsive actuation using a conductive silver paste/Kapton/PDMS composite. The actuator achieves a Joule heating-driven bending angle of approximately 200° under a low operating voltage of 2.0 V within 12 seconds. A sinusoidal function model was employed to fit the deformation behavior. As a demonstration, this study presents a programmable control strategy utilizing multiple heating components. The key contribution of this work is the realization of electrothermally programmable actuation with a low driving voltage and rapid response time, enabled by a simple structure composed of flexible materials. The reviewer provides the following comments:
- The figure captions in Figure 1(a) appear to be incorrect. The positions of “Kapton” and “silver paste” are reversed.
- The font size in Figure 2 is too small.
- In Figure 2(d), what is the cause of the steep change observed in the curve, particularly at an operating voltage of 2.0 V?
- The reviewer believes that the comparison in Figure 2(f) is not appropriate, as the actuators in the referenced studies have different structures from the one presented in this manuscript.
- The flexible actuator was fabricated with a silver paste/Kapton/PDMS structure. Since the key parameters influencing the deformation of the actuators include the thickness of each layer and the overall structure, could the authors provide the thickness of each material?
- The manuscript suggests that electrothermal deformation depends on heat dissipation. How would encapsulating the top layer (silver paste) with an additional PDMS layer affect performance? Specifically, could a sandwiched structure (PDMS/silver paste/Kapton/PDMS) achieve a comparable level of actuation to the original design proposed in this study?
- Safety concerns: Electrothermal actuation is promising; however, the operating temperature is relatively high (~200 °C). Could the authors provide a trade-off analysis or comparative results regarding heat generation and the corresponding actuation performance?
- In Figure 7a, what information can be derived from (iv)? What was the applied driving voltage? Additionally, incorporating temperature change data (e.g., infrared imaging) for Figure 7 could significantly enhance the discussion.
- Could the authors provide insights into the potential of ‘microstructuring’ for the flexible electrothermal actuator? Microstructuring may enable more diverse and complex actuation behaviors while maintaining a reasonable operating voltage.
- Several typographical errors and incorrect spacing between word and word were appeared, such as “PEDOT:PASS” and “Ampulitude”.
Author Response
Response to Reviewers
We sincerely appreciate the reviewers’ constructive comments and suggestions, which have greatly helped us improve the quality and clarity of our manuscript. Below are our point-by-point responses to each of the reviewer’s comments. All changes have been marked in the revised manuscript.
Reviewer Comments and Authors’ Responses
Comment 1: The figure captions in Figure 1(a) appear to be incorrect. The positions of “Kapton” and “silver paste” are reversed.
Response: Thank you for pointing this out. We have corrected the positions of “Kapton” and “silver paste” in Figure 1(a).
Comment 2: The font size in Figure 2 is too small.
Response: Thank you for pointing this out. We have increased the font size in Figure 2 to improve clarity and readability.
Comment 3: In Figure 2(d), what is the cause of the steep change observed in the curve, particularly at an operating voltage of 2.0 V?
Response: We sincerely thank the reviewer for this insightful comment. The original version of Figure 2(d) presented the continuous temperature change at a fixed measurement point on the device surface. However, as the applied voltage increases, Joule heating raises the temperature, causing the device to bend. This bending shifts the relative position of the thermal measurement point with respect to the heating area, leading to an abrupt change in the recorded temperature—most notably around 2.0 V. To eliminate the measurement error caused by deformation-induced displacement, we have replaced the original figure with a revised version that records the maximum temperature across the entire device at each voltage step. This approach provides a more accurate and stable representation of the thermal response. The temperature inflection point in the curve corresponds to the moment when the power is turned off after the actuator reaches a stable bending state.
Figure1 Please see the attachment.
Comment 4: The reviewer believes that the comparison in Figure 2(f) is not appropriate, as the actuators in the referenced studies have different structures from the one presented in this manuscript.
Response: We thank the reviewer for this question. We acknowledge the structural differences in the previous comparison and have revised Figure 2f. We restructured the comparison to focus on: Voltage efficiency (V/deg).
Comment 5: The flexible actuator was fabricated with a silver paste/Kapton/PDMS structure. Since the key parameters influencing the deformation of the actuators include the thickness of each layer and the overall structure, could the authors provide the thickness of each material?
Response: We thank the reviewer for raising this question. We added the thickness values ​​of each layer in lines 139-142 of Section 3.2 and lines 298-301 of Section 3.3: PDMS thickness is approximately 200 μm, Kapton thickness is 50 μm, and silver paste thickness is approximately 100 μm.
Comment 6: The manuscript suggests that electrothermal deformation depends on heat dissipation. How would encapsulating the top layer (silver paste) with an additional PDMS layer affect performance? Specifically, could a sandwiched structure (PDMS/silver paste/Kapton/PDMS) achieve a comparable level of actuation to the original design proposed in this study?
Response: Thank you for your suggestion. We have fabricated a sandwich PDMS/silver paste/Kapton/PDMS structure. This configuration offers improved mechanical protection and more uniform heat dissipation. However, under a 2V DC voltage, the structure shows minimal bending deformation compared to the original silver paste/Kapton/PDMS design (as demonstrated in Figure 2a and b). Additionally, the device's peak temperature is significantly lower (Figure 2c), suggesting that encapsulation impairs thermal concentration and thus reduces actuation efficiency.
Figure 2 Please see the attachment.
Comment 7: Safety concerns due to the relatively high operating temperature (~200°C). Could the authors provide a trade-off analysis?
Response: We appreciate the reviewer’s concern regarding the safety implications of the actuator’s relatively high operating temperature (~200 °C). Although the local maximum temperature can reach approximately 200 °C, it is confined to a small heating region and dissipates rapidly after the power is turned off, ensuring that the overall device remains within a thermally safe operating range. The independently addressable heating units enable localized thermal control, which minimizes the risk of global overheating and enhances operational safety. Moreover, the actuator is primarily intended for industrial applications such as space-deformable antennas in the future, rather than for direct integration with wearable electronics. In such use cases, localized high-temperature actuation is acceptable, especially considering the actuator’s ability to achieve rapid and large deformations under low driving voltage. In future work, we plan to incorporate closed-loop temperature feedback control and microstructured heat dissipation strategies to further improve thermal management and ensure safe, stable operation in more complex environments.
Comment 8: In Figure 7a, what information can be derived from (iv)? What was the applied driving voltage? Additionally, incorporating temperature change data (e.g., infrared imaging) for Figure 7 could significantly enhance the discussion.
Response: We observed from Figure 7a (iv) that each 'petal' bends upward synchronously, demonstrating the uniform deformation of the actuator. This collective actuation further induces an upward bending of the central region, where the antenna is situated. The resulting change in geometry may influence the antenna’s spatial orientation and radiation characteristics. Although not explored in detail in this work, such deformation-coupled reconfiguration suggests potential applications in adaptive or reconfigurable wireless communication systems. We placed our additions in Section 3.4, lines 307-313. The applied DC driving voltage in this experiment was 6.0 V, which was chosen based on preliminary testing to ensure sufficient bending while avoiding thermal damage to the actuator. To further enhance the discussion on thermal effects, we have incorporated infrared imaging data (shown in Figure 7) that provides a clear thermal profile of the actuator during operation. In the thermal image, the highest temperature point is observed at the wire connection area. Point Sp1 represents the highest temperature within the device region, excluding the wiring, and thus more accurately reflects the thermal state of the functional structure.
Comment 9: Could the authors provide insights into the potential of ‘microstructuring’ for the flexible electrothermal actuator? Microstructuring may enable more diverse and complex actuation behaviors while maintaining a reasonable operating voltage.
Response: We sincerely appreciate the reviewer’s insightful comment regarding the potential of microstructuring. We fully agree that microstructuring is a highly promising strategy for enhancing the actuation performance of flexible electrothermal actuators. In this study, we have primarily focused on achieving multi-mode programmable deformation through optimization of the conductive silver paste patterns and independent control of heating units. However, we recognize that incorporating periodic or complex microstructures—either on the heating layer or within the PDMS substrate—could further improve heat distribution, reduce localized thermal accumulation, and enable more precise localized actuation. These advantages would facilitate the realization of more diverse and sophisticated deformation behaviors under low-voltage operation. Although this aspect has not yet been addressed in our current work, we consider it a valuable direction for future research. We plan to explore the integration of laser etching and high-resolution flexible printing techniques to implement microstructuring strategies and systematically study their impact on actuator performance. This will contribute to advancing intelligent control and multifunctional capabilities in flexible actuator systems.
Comment 10: Several typographical errors and incorrect spacing between word and word were appeared, such as “PEDOT:PASS” and “Ampulitude”.
Response: Thank you for pointing this out. We have carefully checked and corrected all typographical errors and spacing issues throughout the manuscript, including those mentioned.
We sincerely appreciate the reviewers' insightful comments and valuable suggestions.

Round 2
Reviewer 2 Report
Comments and Suggestions for Authors
The revised manuscript mostly addressed the questions raised by the reviewer. Thus, I recommend that this work should be published after careful proofreading to correct the minor errors in English.